# Rho GTPases in Skeletal Muscle Development and Homeostasis

**DOI:** 10.3390/cells10112984

**Published:** 2021-11-02

**Authors:** Sonia Rodríguez-Fdez, Xosé R. Bustelo

**Affiliations:** 1Molecular Mechanisms of Cancer Program, Centro de Investigación del Cáncer, CSIC-University of Salamanca, 37007 Salamanca, Spain; xbustelo@usal.es; 2Instituto de Biología Molecular y Celular del Cáncer, CSIC-University of Salamanca, 37007 Salamanca, Spain; 3Centro de Investigación Biomédica en Red de Cáncer (CIBERONC), CSIC-University of Salamanca, 37007 Salamanca, Spain; 4Wellcome-MRC Institute of Metabolic Science and MRC Metabolic Diseases Unit, University of Cambridge, Cambridge CB2 0QQ, UK

**Keywords:** small G protein, RhoA, Rac1, Cdc42, guanosine nucleotide exchange factors, GTPase activating proteins, rock, pak, signaling, myogenesis, muscle mass, muscle regeneration, satellite cells, metabolism

## Abstract

Rho guanosine triphosphate hydrolases (GTPases) are molecular switches that cycle between an inactive guanosine diphosphate (GDP)-bound and an active guanosine triphosphate (GTP)-bound state during signal transduction. As such, they regulate a wide range of both cellular and physiological processes. In this review, we will summarize recent work on the role of Rho GTPase-regulated pathways in skeletal muscle development, regeneration, tissue mass homeostatic balance, and metabolism. In addition, we will present current evidence that links the dysregulation of these GTPases with diseases caused by skeletal muscle dysfunction. Overall, this information underscores the critical role of a number of members of the Rho GTPase subfamily in muscle development and the overall metabolic balance of mammalian species.

## 1. Introduction

In addition to being essential for supporting the skeletal structure and locomotion, skeletal muscle plays critical roles in the disposal of glucose in response to insulin, in heat production, and in thermoregulation. Due to those roles, the function of this tissue is also required to maintain metabolic homeostasis at the organismal level [1,2,3]. Skeletal muscle is a very plastic organ that can adapt its size and functionality in response to different external and internal cues. Such factors include muscle cell turnover, stem cell numbers, regeneration capacity, exercise, nutritional status, and age. Muscle mass, strength, and functionality usually decline with age due to a negative protein turnover balance that leads to the concurrent degeneration of muscle fibers and a decrease in the regenerative capacity of the tissue [4]. Loss of muscle homeostasis can also occur in earlier life periods, leading to the development of musculoskeletal disorders [5,6,7].

Skeletal muscle fibers are multinucleated syncytia formed by the fusion of multiple cells in a developmental process called myogenesis. The primary fibers of skeletal muscle arise from the original somite cells. These are differentiated in several waves during embryonic myogenesis to shape the basic muscle architecture plan. After this process, secondary fibers are generated and added to the already formed myofibers during the process referred to as fetal differentiation (Figure 1) [8]. Late in embryogenesis, a small population of cells with self-renewing and differentiating capacities adopt a satellite position with respect to myofibers. These stem cells, which are maintained throughout the life of healthy individuals, are responsible for the growth of the muscle during youth as well as for muscle regeneration [9]. In adults, satellite cells are dormant under normal conditions. However, they can exit quiescence and acquire an active state upon specific stimuli such as stimulation by growth factors or physical trauma in a process that recapitulates the steps of embryonic and fetal myogenesis (Figure 1) [9]. In all cases, the committed precursors differentiate into proliferating myoblasts that, eventually, will exit cell cycle to undergo further differentiation steps and generate fusion-competent myocytes. Prior to their fusion, myocytes must adopt an appropriate position to allow cell-cell adhesion and the alignment of their membranes, processes that require complex cytoskeleton rearrangements [10]. The fusion of these cells ultimately originates the multinucleated syncytial myotubes that will mature to form contractile myofibers (Figure 1) [9,11]. Muscle mass can increase postnatally through either the addition of new muscle fibers or to the hypertrophy of pre-existing ones, the last option being the most common in adult muscle. Exercise or anabolic stimulation can lead to an increase in the number of intracellular proteins and organelles, causing the expansion of the muscle fibers or hypertrophy [6]. Catabolic conditions trigger opposite effects [6,12]. The regulation of these processes can be mediated by local and systemic signals, such as catecholamines, hormones (i.e., the hypothalamic growth hormone or insulin-like growth factor-1) and mechanical stimulation [13,14,15].

The canonical Rho GTPases work as molecular switches during cell signaling [16]. These proteins have been classically considered key players in the regulation of the F-actin cytoskeleton and associated biological processes such as cell adhesion, migration, and polarity. However, it is now known that they participate in many other cellular processes such as the regulation of microtubule dynamics, cell cycle progression, transcriptional regulation, and vesicle trafficking [17,18,19,20]. Their roles in the regulation of physiological and pathological processes are equally wide. For example, it is known that Rho GTPases coordinate the differentiation of several cell types and that they play crucial roles in the regulation of immunological responses, blood pressure levels, and glucose homeostasis [21,22,23,24,25,26,27,28,29,30,31,32]. As a result, alterations in their activity due to mutations, changes in expression and the deregulation of upstream and downstream signals can lead to a number of diseases such as cancer or cardiovascular, neurological, autoimmune, and immunodeficiency disorders [28,33,34]. Recent work has also revealed that Rho GTPases play critical roles during muscle development, regeneration, and function. In this review, we will address their intrinsic roles in those processes that have been discovered during these last years. Prior reviews on some of these aspects can be found elsewhere [10,24,26,32,35,36].

## 2. Rho GTPases Regulation, Family Members, and Roles

The Rho GTPase family is composed of 20 proteins. Most of them exhibit an almost ubiquitous expression, although some (e.g., RhoH, Rac2) show tissue-specific distribution [37]. According to structural homology criteria, Rho GTPases can be subdivided in six subfamilies. Out of those, the RhoA, Rac1 and Cdc42 subfamilies are the best studied so far both at the cellular and organismal level [20]. In order to carry out properly regulated functions in cells, Rho GTPases are subjected to a very tight spatiotemporal control that involves many upstream and downstream regulators (Figure 2). It is worth mentioning, however, that there is a subset of atypical Rho GTPases (e.g., RhoE) characterized by being present in a constitutive active state [38,39]. The activation of Rho GTPases that do cycle between GDP and GTP states is mediated by a large group of enzymes known as GDP/GTP exchange factors (GEFs) [40]. These proteins catalyze the release of the bound GDP molecules that, in turn, favors the incorporation of the GTP molecules in the guanosine nucleotide binding site of the GTPases (Figure 2). Depending on the catalytic domain, these GEFs can belong to the so called Dbl-homology and Dedicator of Cytokinesis (Dock)-homology protein families. As we will see in this review, some of these proteins have been involved in a number of skeletal muscle-specific functions (ArhGEF3, Dock1, Dock5, Trio, ArhGEF25, Vav2, Obscurin) [41,42,43,44,45,46,47,48]. Upon activation, the GTP-bound GTPases become anchored to the plasma membrane via its C-terminal prenylated tail [49]. Other C-terminal polybasic amino acid sequences and/or palmitoylation events can cooperate in this step [50]. The subsequent inactivation step is at the hands of the GTPase activating proteins (GAPs), which catalyze the hydrolysis of the GTP molecules bound to the active versions of the GTPases (Figure 2) [51]. GAPs involved in skeletal muscle include ArhGAP5, ArhGAP26 and ArhGAP35 [52,53,54,55]. A third regulatory layer includes the binding of the GDP-bound versions of Rho GTPases to guanine nucleotide dissociation inhibitors (GDIs) (Figure 2) [51]. This interaction maintains the Rho GTPases arrested in the cytosol due to sequestration of the geranyl-geranyl group of Rho GTPases by a hydrophobic groove located within the 3D structure of Rho GDIs [56] (Figure 2). This interaction also allows the long-term stability of Rho GTPases since the elimination of Rho GDIs leads to the rapid degradation of those GTPases in cells [56]. The Rho GTPase–Rho GDI interaction can be further regulated by phosphorylation events (e.g., Src- or Pak-dependent) or by interactions of the complex with cytoskeletal proteins such as coronin [57,58,59]. To date, 84 Rho GEFs, 66 Rho GAPs, and 3 Rho GDIs have been identified in humans [51,60]. Additional regulatory layers found in Rho GTPases include changes in expression, cytoskeletal-mediated nanoclustering events, and a large variety of posttranslational modifications such as phosphorylation, ubiquitination, and SUMOylation. Further information on these regulatory steps can be found in other review articles [20,51,61,62].

Active Rho GTPases can interact with a large spectrum of downstream effectors (Figure 2). Up to now, more than 200 effectors have been identified [63]. Those include kinases (e.g., Rho kinase [ROCK] and p21-activated kinase [PAK)] family members) and scaffolding proteins such as the mammalian homolog of *Drosophila* diaphanous 1 (mDia1) and the Wiskott-Aldrich syndrome protein (WASP) (Figure 2). Some of those effectors show specificity for particular Rho GTPase family members. For example, GTP-bound RhoA binds to the proximal effectors mDia1 and ROCK to favor the formation of both stress fibers and focal adhesions [64]. Likewise, GTP-bound Rac1 can bind to the WASP family verprolin homologous protein (WAVE) to favor lamellipodia formation in cells [65]. Other effectors, however, show less specific binding patterns towards the upstream GTPases. For example, GTP-bound Rac1 and Cdc42 can interact with PAK family kinases (Figure 2) [66]. More distal downstream elements of activated Rho GTPases include additional kinases (e.g., c-Jun N-terminal kinase [JNK]) as well as transcriptional factors such as serum responsive factor (SRF), the nuclear factor Κ B (NFΚB), and the Yes-associated protein (YAP)/WW domain containing transcription regulator 1 (TAZ) complex [51,67,68,69,70,71,72]. As we will discuss below, some of these upstream and downstream factors play critical roles in skeletal muscle functions.

### 2.1. Rho GTPases in Muscle Differentiation

Previous studies have shown that Rho GTPases are relevant for both embryonic and adult myogenesis, periods in which they regulate processes linked to both cell fate specification and late skeletal myogenesis steps. These functions are very complex, since different members of the Rho family can concurrently act in a concerted manner and, in some other cases, play opposite roles at specific differentiation steps. RhoA is perhaps the most important player in those processes. Thus, it has been shown that it can contribute downstream of the insulin growth factor 1 (IGF1) to the commitment of muscle precursor cells by favoring the fate of the mesenchymal stem cells towards the myogenic versus the adipogenic lineage. As a result, the genetic elimination of the ArhGAP5 Rho GAP (also known as p190-B RhoGAP) affects adipogenesis in mice and promotes the spontaneous differentiation of murine embryonic fibroblasts into muscle cells [48,65] (Figure 3A). RhoA is also necessary for the initial induction of the myogenic program via the stimulation of SRF-mediated gene expression programs and, subsequently, to maintain the myoblasts in a proliferative state [73,74,75] (Figure 3A). Later on, the activity of this GTPase must be shut down to prompt cell cycle withdrawal and the final myoblast fusion step [53,75,76,77]. This latter step is mediated by the combined action of the Rho GAPs ArhGAP26 and ArhGAP35 [53,55] (Figure 3A). In the latter case, the activation of this GAP seems to require its prior binding to RhoE [55]. Consistent with this negative regulatory role, cell culture experiments have shown that RhoA fluctuates between high and low activity states in nondifferentiated and differentiating myoblasts, respectively (Figure 3B) [75,77,78]. However, it remains to be determined whether such fluctuations also occur in vivo. Finally, it has been shown that RhoA activity regulates the switch between embryonic and fetal myogenesis in mice. To achieve this, the RhoA–ROCK axis maintains the cells in embryonic myogenesis by indirectly repressing JunB and Nuclear Factor One X (NFIX). The latter one is a transcription factor involved in this embryonic to fetal transcriptional switch checkpoint [79,80,81]. Consistent with such roles, the activity of RhoA drops at the onset of fetal embryogenesis in mice. Likewise, the inhibition of ROCK in embryonic myoblasts leads to a marked increase in NFIX levels [79]. Although NFIX also plays roles in adult muscle regeneration, it seems that such a role is RhoA-independent [79,82].

Rac1 and Cdc42 seem to play roles at late stages of the differentiation of skeletal muscle. Consistent with this, it has been observed in cell culture that the levels of active Rac1 and Cdc42 increase during the differentiation phase of myoblasts (Figure 3B) [44,78]. This pattern of expression is opposite to that observed in the case of RhoA (Figure 3B) [75,77,78]. In line with this, it has been shown that Rac1, Cdc42 and their effector proteins Pak1 and Pak3 are required for myoblast fusion in mice (Figure 3A) [44,83,84,85]. In the case of flies and zebrafish, genetic analyses indicate that the activation of these proteins prior to myoblast fusion is mediated by the orthologous proteins for Dock1 and Dock5 GEFs [43,86] (Figure 3B). The function of Dock1 is phylogenetically conserved in mice [42,43,87]. Further studies indicate that Dock1 activation is triggered by the engagement of BAI3 receptor signaling [88,89] (Figure 3B). Finally, the Rho GEF Trio is involved in the activation of those GTPases downstream of M-Cadherin during the myoblast fusion step. The myogenic defects in *Trio^–/–^* mice are different and milder than those observed in *Dock1^–/–^* animals [42], suggesting Trio is only required for the fusion phase during secondary or fetal myogenesis [44,45] (Figure 3B). Using cellular models, it has also been shown that the ectopic overexpression of the Rac1 GEF Def6 in a muscle cell line promotes the fusion of myoblasts as well [90]. However, and in contrast to the phenotypes exhibited by Trio and Dock1 deficient mice [42,44,45], this function has not been corroborated using in vivo models as of yet. It would be important to investigate whether this lack of phenotype is due to redundancy issues with those GEFs using compound mice lacking Def6 and each of the other two Rho GEFs.

### 2.2. Rho GTPases in Muscle Regeneration

The regenerative capacity of skeletal muscle is crucial for the homeostatic regulation of the tissue. An impairment in this process or in the number of satellite cells may lead to the development of severe and incapacitating diseases [91]. The dormant pool of satellite cells is located peripherally to the fibers and under the basal lamina (Figure 4). The cells lie flat until they are activated by injury signals or other stimuli. They then re-enter cell cycle and migrate to the damaged region [91]. Recent observations indicate that RhoA controls this process by two independent mechanisms. On the one hand, it has been observed that, in absence of damage, satellite cells maintain RhoA in an active state through stimulation by the Wnt4 ligand secreted by muscle cells. This activation is thought to occur through a non-canonical Wnt signaling pathway, first identified during gastrulation, that requires the Frizzled/Dishevelled 2-mediated stimulation of Dishevelled-associated activator of morphogenesis 1 (Daam1) [92,93]. RhoA activation, in turn, promotes the inhibition of YAP in a ROCK-dependent mechanism [92]. Since YAP is key for breaking quiescence, the active Wnt4–RhoA–ROCK axis favors the maintenance of the satellite cells in a resting state (Figure 4) [94]. This signaling pathway is silenced via the repression of Wnt4 synthesis upon muscle injury, leading to the stimulation of YAP and the transition of the satellite cells to an active state [92,95]. On the other hand, it has been shown that RhoA regulates the autophagy flux during muscle regeneration. Autophagy is emerging as a key regulator of muscle regeneration that can also provide the extra burst of energy required to switch from the quiescent to the activated state in satellite cells [96]. Consistent with this role, a recent study has shown that the inactivation of the RhoA–ROCK axis caused by the depletion of the upstream exchange factor ArhGEF3 promotes injury-induced muscle regeneration by increasing autophagy in mice (Figure 4) [41]. 

Several lanes of evidence also suggest that the Rho GEF ArhGEF25 is probably involved in this process. For example, its mRNA has been found to be upregulated during muscle regeneration. Furthermore, its ectopic expression promotes muscle regeneration in mouse models of muscle injury in a Rho GTPase-dependent manner [97]. Although no further characterization has been performed, its expression pattern suggests that this Rho GEF is involved in the latest steps of the regenerative process [97].

Interestingly, the analysis of compound, skeletal muscle-specific *Pak1^–/–^;Pak2^–/–^* knockout mice revealed the progressive development of a late-onset megaconial myopathy-like condition, indicating that Rac1 and/or Cdc42 also influence skeletal muscle regeneration. The cause of this phenotype is unknown as yet [98]. It is important to note, however, that these two serine/threonine kinases have been involved in the proper timing of myoblast differentiation using similar genetic analyses [85].

Rho GTPases also control muscle regeneration using satellite cell extrinsic mechanisms. For example, it is known that the proper activation and subsequent differentiation of satellite cells requires the elimination of the damaged muscle fibers from the injured area. The elimination of the cell debris also requires inflammatory signals from infiltrated macrophages [99]. The silencing of the RhoA–ROCK axis in those inflammatory cells induces phagocytosis, and this is an important step during this post-injury phagocytic stage [100].

### 2.3. Rho GTPases in Muscle Mass Regulation

Skeletal muscle mass homoeostasis is achieved through the adequate balance between anabolic and catabolic processes taking place in muscle fibers. Increased rates of muscle mass production occur when the overall rates of protein synthesis are higher than degradation. Conversely, loss of muscle mass or atrophy is a sign of increased protein degradation in the fibers [7]. This latter process is quite apparent under poor nutrition conditions and during prolonged inactivity states, chronic and neuromuscular diseases, cancer-associated cachexia, and aging [7,12]. Muscle hypertrophy can be driven by growth factors, hormones or in response to mechanical cues during exercise. One of the main signaling pathways regulating this process is the IGF1-Akt-mTOR pathway [7]. We have recently shown that the expression of a catalytically hyperactive or a hypomorphic Vav2 mutant version from the endogenous Vav2 locus promotes increased and reduced lean mass, respectively, in knock-in mouse models [46]. Further experiments indicate that the Rho GEF activity of Vav2 contributes to the activation of the IGF1 signaling pathway, being Rac1 the Rho GTPase activated by Vav2 in this new regulatory layer of the IGF1-Akt-mTOR pathway [46]. Mice deficient in both Pak1 and Pak2, two of the main Rac1 and Cdc42 effectors, also show reduced muscle mass and fibers with smaller cross-sectional area [85]. However, unlike the case of the Vav2 knock-in mouse models, these defects are associated with reduced numbers of muscle cells rather than to the proper regulation of the signaling output from the IGF1 receptor [46,85]. This is consistent with the observation that, in the case of the Vav2 pathway, Rac1 used an F-actin-dependent but a Pak-independent mechanism to promote the optimal activation of the IGF1 receptor–Akt pathway [46]. In the case of Pak proteins, this phenotype can be also connected to the role of these kinases in myogenesis and muscle regeneration described above [84,85,101]. Additional information on other possible roles of Rho GTPases in the regulation of muscle mass can be found in a previous review [27]. 

### 2.4. Rho GTPases Signalling in Sarcomere Banding Patterning

Sarcomeres are formed by F-actin and myosin filaments in a specific band configuration that provides contractile capacity to the muscle. The organization of the sarcomeres and establishment of this banded pattern is called myofibrillogenesis [102]. One of the structural components of muscle with a role in this process is obscurin, a giant sarcomeric Rho GEF [47]. Using human primary myoblasts differentiated in vitro, it has been shown that this GEF promotes myofibrillogenesis via the stimulation of the Rho GTPase TC10 [103]. This regulatory step is probably quite relevant, as it has also been found in *Ascidia* [104]. It is possible that obscurin could stimulate RhoA in this process too, as they both colocalize at the M-bands. In addition, it has been shown that obscurin can stimulate RhoA when using a contraction-induced muscle injury model in rats [48]. Consistently, obscurin is now considered a potential therapeutic target in skeletal muscle and cardiac disorders [105,106].

### 2.5. Rho GTPases in Skeletal Muscle-Related Diseases

Skeletal muscle is the main organ of the body that contributes to the uptake of glucose induced by insulin [1,2]. Given its prominent role and extension, alterations in muscle mass can affect metabolic homeostasis as it has been extensively demonstrated in mice and humans [107,108,109,110,111,112,113]. The promotion of muscle hypertrophic pathways through inhibition of myostatin (a secreted growth factor that negatively regulates the skeletal muscle mass) or induction of IGF-1 in muscle both can lead to an improvement in insulin sensitivity and other metabolic parameters [107,108,109,110]. Conversely, such parameters are negatively affected in cases associated with reduced skeletal muscle mass. The phenotype of loss- and gain-of-function knock-in mouse models for the Vav2 GEF illustrates well this connection between lean mass and proper metabolic balance [46]. Thus, mice expressing a catalytically hypoactive version of Vav2 (L332A mutant) show a progressive reduction in skeletal muscle mass and, in parallel, the subsequent development of adiposity in both the brown and white adipose tissue. This, in turn, leads to the generation of both liver steatosis and hyperglycemia with the aging of the animals [46]. These problems are further accelerated and aggravated when those mice are subjected to a high-fat diet [46]. By contrast, mice expressing a catalytically hyperactive version of Vav2 show increased muscle mass and resistance against all the foregoing metabolic dysfunctions even when fed with a high-fat diet [46]. A detailed description of the role of Vav2 in skeletal muscle and associated metabolic process can be found in a recent review [32]. In line with this, several clinical studies have established links between reductions in lean mass and type II diabetes [111,113].

Type II diabetes can also originate as a consequence of the development of insulin resistance by muscle fiber cells [114]. The Rac1 GTPase is crucial in this process, as it mediates the translocation of the glucose transporter Glut4 to the plasma membrane in insulin-stimulated cells [36,115,116,117,118]. In support of the involvement of Rac1 in this process, a recent study has shown that type II diabetic patients usually show low levels of Rac1 activation in skeletal muscle upon insulin stimulation [118]. The Rho GEF Plekhg4 has been shown to promote Rac1 activation in this process [119]. Rac1 and the Rho GEFs Tiam1 and Kalirin have also been shown to mediate muscle glucose uptake in response to other stimuli [120,121,122,123,124]. It has also been posited that the RhoA–ROCK1 axis can positively influence insulin signaling in muscle and adipose cells through phosphorylation of the insulin receptor substrate 1 [125,126,127]. It is worth noting, however, that these results are controversial since muscle-specific knock-in mice expressing a constitutively active version of ROCK1 become insulin resistant [128]. These and other functions of Rho GTPases in the regulation of glucose homeostasis have already been extensively reviewed elsewhere [26,27].

A second group of diseases includes muscular dystrophies, a group of genetic diseases characterized by progressive weakness and degeneration of skeletal muscle. These diseases are quite diverse in terms of pathological impact, time of onset, and degree of severity. One of the most common and severe types of those diseases is Duchenne muscle dystrophy, which is caused by mutations in *DMD* [129]. This gene encodes a protein called dystrophin that forms complexes with extracellular and transmembrane glycoproteins. These complexes anchor the cytoskeleton to the extracellular matrix, thus protecting the sarcolemma from contraction-induced stress and injury [129]. Rho GTPases have not been directly linked so far to the onset of muscular dystrophy. However, it is known that both Rac1 and Cdc42 are part of the dystrophin-glycoprotein complex [130,131]. Moreover, mouse experiments indicate that disrupting Cdc42 signaling in this context impairs cell-to-cell contacts [130]. Consistent with this hypothesis, it has been found that the elimination of the Rho GAP ArhGAP26 (also known as GRAF1, Ophn1L and KIAA0621) in mice leads to delayed membrane resealing following acute sarcolemma damage. This phenotype is further exacerbated when *ArhGAP26* is deleted in dystrophin-deficient mice [132]. Collectively, this evidence suggests that alterations in the signaling of these GTPases could contribute to the pathological effects elicited by DMD mutations in skeletal muscle fibers. Supporting this idea, it has been found that an intronic variant of *DOCK1*, a gene encoding a Rac1 GEF, seems to correlate with the age in which the ambulatory capacity of Duchenne muscle dystrophy patients is lost [133]. Likewise, it has been reported that the use of a ROCK inhibitor (Y-27632) in combination with the gold standard treatment of corticosteroids ameliorates the Duchenne muscle dystrophy-like condition in mice [129,134,135].

## 3. Concluding Remarks

Rho GTPases play important roles in muscle development, regeneration, and homeostasis. However, many aspects remain to be clarified as of yet. For example, the mechanisms by which the Rho GTPase-specific pathways are switched on and off during myogenesis is not clear. The level of conservation of these pathways in embryonic, fetal, and adult periods is also unclear. Independently on the answer to this latter question, we still have to dissect all the signaling machinery involved in each of those processes and to discern how it dynamically fluctuates during the course of all of those phases. One of the limitations in the Rho GTPase field is the existence of multiple redundant and/or compensatory layers that can impact the signaling output of a given biological process. This is mainly due to the high number of GEFs, GAPs, and effectors that can sometimes cover for the loss of other regulators or act as a dual safety mechanism to ensure physiological homeostasis. In the same context, the physiological interconnections that exist among skeletal muscle, brain and brown and white adipose tissue further complicate the identification of skeletal muscle-intrinsic effects of the dysfunctions that are observed in vivo. The use of more sophisticated animal models (e.g., inducible, skeletal muscle-specific) may help to better define the role of Rho GTPases and identify their regulators and effector pathways in these processes. The full dissection of all these Rho GTPase-regulated biological programs can be of interest not only for the basic understanding of skeletal muscle biology but, likely, as a way to develop new therapies for diseases associated with skeletal muscle diseases. The preliminary results indicating that a ROCK inhibitor can be of interest for DMD underscores the potential interest of this idea [134].

Another aspect that is hitherto ill-known is the role of the rest of members of the Rho GTPase family that have not been as intensely studied as the classical Rac1, RhoA and Cdc42 proteins. The involvement of TC10 in human myofibril organization [103] is a clear demonstration that the study of this largely neglected subset of Rho family proteins is worth taking. Arguably, further work on the role of Rho GTPases, regulators, and pathway elements must shed further light in the development, maintenance, and function of skeletal muscle.

## Figures and Tables

**Figure 1 cells-10-02984-f001:**
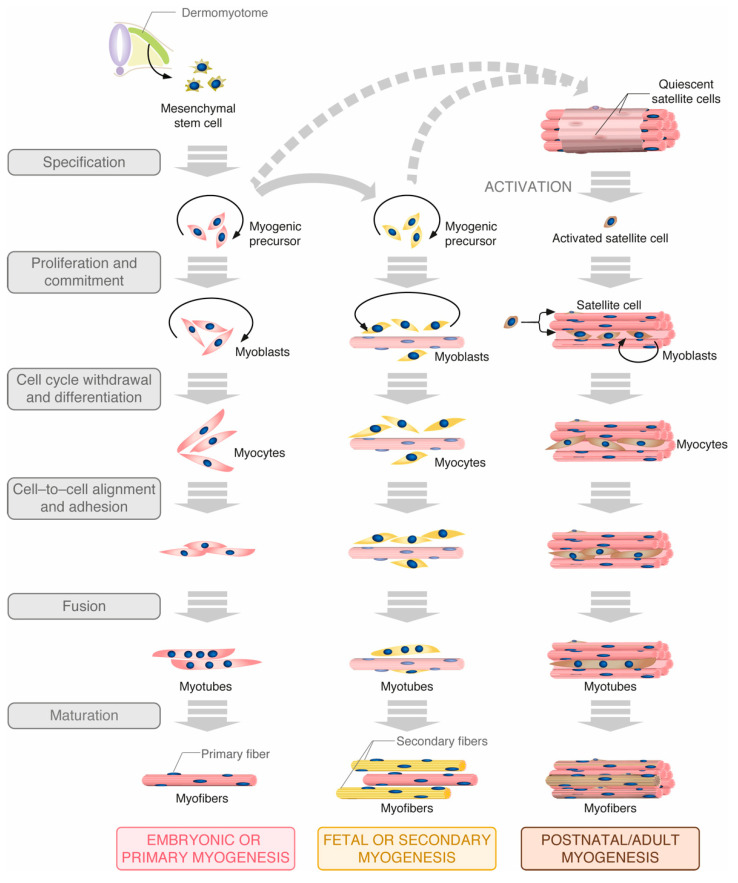
Main steps of embryonic, fetal and postnatal myogenesis. Proliferation and cell division stages are indicated by black arrows. See further details in main text.

**Figure 2 cells-10-02984-f002:**
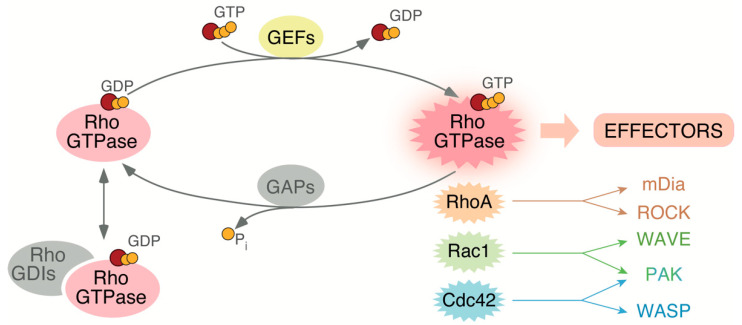
Rho GTPases activation cycle is tightly regulated. Rho GTPases remain inactive at the cytoplasm in their GDP–bound state (**left**) and become active upon binding to GTP (**right**). GEF proteins can bind to the switch domain of the GTPases to favor the exchange of GDP for GTP to promote diverse specific responses both at the cellular and organismal level. Hydrolysis of the GTP to GDP stimulated by GAPs returns the Rho protein to the inactive state. GDIs can sequester the inactive protein in the cytoplasm preventing its translocation to the membrane and its activation by GEFs. The most studied Rho GTPase family members and their main effectors are depicted in the right. Abbreviations have been described in the main text.

**Figure 3 cells-10-02984-f003:**
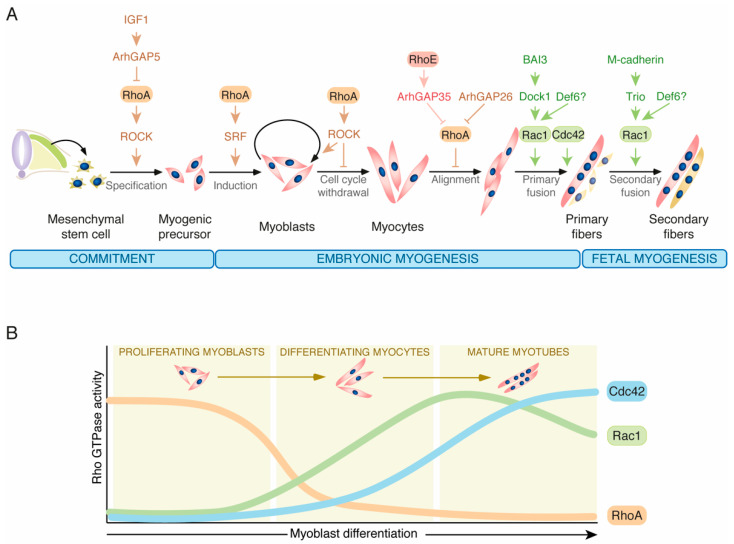
Known roles of Rho GTPases during myogenesis. (**A**) Roles of Rho GTPases during embryonic (primary) and fetal (secondary) myogenesis according to current knowledge. Activation steps are indicated by arrows and inhibitory roles are indicated by blunted lines. Abbreviations are defined in the main text. (**B**) Pattern of activation of Rho GTPases along differentiation according to published data from in vitro experiments using mainly the C2C12 murine myoblast cell line.

**Figure 4 cells-10-02984-f004:**
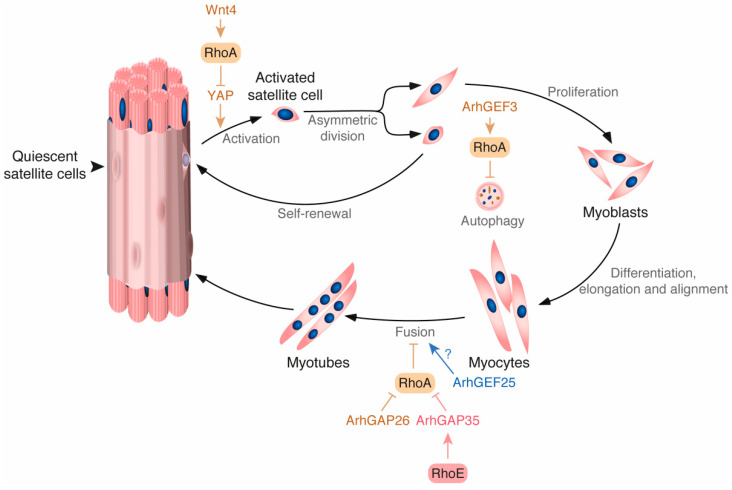
Rho-GTPase regulated pathways and processes during muscle regeneration. Activation and inhibitory steps are indicated by arrows and blunted lines, respectively. Abbreviations used have been described in the main text.

## Data Availability

Not applicable.

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
