# Peer review of "Rho GTPases in Skeletal Muscle Development and Homeostasis"

_cells, 2021, doi:10.3390/cells10112984_

Round 1

Reviewer 1 Report

Rho GTPases in Skeletal Muscle Development and homeostasis

Sonia Rodríguez-Fdez and Xosé R. Bustelo

This review summarizes the current knowledge on the role of Rho signaling pathways in skeletal muscle development, regeneration, mass homeostasis, metabolism, as well as in diseases caused by skeletal muscle dysfunction.

This is an interesting review, well focused and clearly organized into five main sections: i) Rho GTPases in muscle differentiation; ii) Rho GTPases in muscle regeneration; iii) Rho GTPases in muscle mass regulation; iv) Rho GTPases signalling in sarcomere banding patterning; v) Rho GTPases in skeletal muscle-related diseases.

However, although clearly written, this review could be improved in two ways: it should cite original references as much as possible and describe what is currently known on several aspects with more accuracy, as described in the following points.

  1. Introduction:

- Li 67: "Ras-like as molecular switches". Please cite the original work of Bourne, Sanders and McCormick, published in 1992, much earlier than the proposed references (10.1038/348125a0)

  1. Rho GTPases regulation, family members, and roles.

- Li 87-88: " The Rho GTPase family is composed of 21 proteins with almost ubiquitous expression ..."

The Rho family of GTPases is actually made of 20 proteins, not 21. This was established by Boureux et al. 2007 (DOI: 10.1093/molbev/msl145). In this reference, it is also established that nine of the 20 members show a marked tissue-specific expression. The sentence should be modified accordingly and the reference should be added.

- Li 93-94: The sentence "a small subset of Rho GTPases (e.g., RhoE) is characterized by being present in a constitutive active state" is not correct, since at least eight Rho members (i.e. 40%, not a "small subset") are either devoid of GTPase activity or exchange spontaneously GDP with GTP. This has been reviewed in Aspenström et al. 2007 (10.1016/j.yexcr.2007.07.022). The sentence should be modified accordingly and the reference should be added.

- Li 101: " skeletal muscle-specific functions (ArhGEF3, Dock1, Dock5, Trio, Vav2)

The authors should also add to the list Obscurin, involved in skeletal muscle functions, as described li 263-272, and ARHGEF25/GEFT (Bryan et al. 2005, 10.1128/MCB.25.24.11089-11101.2005).

- Li 115-116: The authors cite the ref #45 to state that the human genome encodes 85 RhoGEFs. However, the number of Dbl-like members had been miscounted in this reference, which stated 74 in the legend whereas only 70 were listed in the table. Since then, the human repertoire of Dbl-like proteins has been worked out extensively in Fort and Blangy, 2017 (DOI: 10.1093/gbe/evx100), which identified 71 Dbl-like genes in the human genome. In addition, SWAP70 (Shinohara et al. 2002, 10.1038/416759a) and SWAP70L/DEF6 (Gupta et al. 2003, 10.1016/s0198-8859(03)00024-7) have been considered as Dbl-like RhoGEFs, although their exchange domains do not have the conserved residues found in the DH domains of other Dbl-like members. The RhoGEF family in humans is thus made of 11 DOCK genes, 71 Dbl-like and 2 SWAP70. The sentence should be modified accordingly and the reference should be added.

- Li 131: The authors indicate that "more than 70 effectors have been identified". As worked out by Bagci et al. 2020 (DOI: 10.1038/s41556-019-0438-7), 239 effectors have been identified in only two human cell types (HeLa and HEK293T cells). The sentence should be modified accordingly and the reference should be added.

2.1 Rho GTPases in muscle differentiation:

- Li 191: Ref #39 (Laurin et al. 2008) is not at the correct place. Since it only concerns Dock1‑/‑ mice, it should be placed at the beginning of the line above, after "animals".

- TC10 should be added to the list of GTPases involved in differentiation, as mentioned in section §2.4.

- DEF-6 also influences myoblast differentiation (Samson et al. 2007, 10.1074/jbc.M611197200). The reference should be added.

2.2 Rho GTPases in muscle regeneration:

In addition to ARHGEF3, the authors should add ARHGEF25, shown to play roles in myogenesis and muscle regeneration (GEFT, Bryan et al.2005, 10.1128/MCB.25.24.11089-11101.2005)

2.3 Rho GTPases in muscle mass regulation:

This section reports on skeletal muscle mass homeostasis, with a particular emphasis on Vav2 knock-in mice (Ref #43), as recently published by the authors.

Since the review published in Cells in 2019 (Ref #26) overlaps some aspects developed in this section with more details, it is needed to state that this section summarizes recent work on Vav2 and specify that the roles of other components of Rho GTPase signaling pathways in exercise and glucose uptake have been reviewed in Ref  #26.

2.4 Rho signalling in sarcomere banding pattern:

Additional information could be included in this very short section:

Benian et al. 1996 (10.1083/jcb.132.5.835 ) identified the presence of DH/PH domains in unc-89, the C. elegans ortholog of obscurin. Young et al. 2001 (ref #94) also showed the presence of the DH/PH domains but did not show that obscurin is a RhoGEF (i.e. capable of GDP/GTP exchange activity). This was done in Qadota et al. 2008 (10.1016/j.jmb.2008.08.083), showing that the DH/PH domain of unc-89 has an exchange activity toward Rho, not Rac, Cdc42 or Mig-2. Note also that O Raeker et al. 2010 (10.1016/j.ydbio.2009.11.018) showed the direct implication of the RhoGEF domain in the physiological functions of obscurin in the zebrafish, by using a morpholino-mediated in-frame deletion approach.

Last, obscurin is considered as a potential therapeutic target in muscle disorders (Randazzo et al. 2017, 10.1080/14728222.2017.1361931; Perry et al. 2013, 10.1002/iub.1157). This can serve as a link between this section and the next one.

References should be added and the text modified accordingly.

2.5 Rho GTPases in skeletal muscle-related disease.

Additional references should be included and the text modified accordingly:

In stretch or contraction stimulated muscles, glucose uptake requires Rac1 activity (Sylow et al. 2014, 10.1113/expphysiol.2014.079194 and Sylow et al. 2015, 10.1113/jphysiol.2014.284281). Tiam1 may be one RhoGEFs responsible for Rac1 activation in this process (Yue et al. 2021, 10.1096/fj.202001312R; You et al. 2013, 10.1016/j.cellsig.2013.08.018).

The RhoGEFs  Sos (Kim et al. 1999, 10.1006/bbrc.1998.9940), Kalirin (Lee et al. 2017, 10.1016/j.cellsig.2016.10.013), SWAP70 (Ueda et al. 2008, 10.1042/BC20070160) have been also involved in glucose uptake.

Last, GRAF1/ARHGAP26 is necessary for efficient muscle membrane repair and may be involved in the pathological progression of muscular dystrophy patients  (Lenhart et al. 2015, 10.1186/s13395-015-0054-6).

Figures:

Figures should be modified according to new references included in the text.

More contrast should be added as some parts are difficult to read.

Author Response

Reviewer comment: This review summarizes the current knowledge on the role of Rho signaling pathways in skeletal muscle development, regeneration, mass homeostasis, metabolism, as well as in diseases caused by skeletal muscle dysfunction.

This is an interesting review, well focused and clearly organized into five main sections: i) Rho GTPases in muscle differentiation; ii) Rho GTPases in muscle regeneration; iii) Rho GTPases in muscle mass regulation; iv) Rho GTPases signalling in sarcomere banding patterning; v) Rho GTPases in skeletal muscle-related diseases.

Answer: Thank you for your kind comments.

Reviewer comment: However, although clearly written, this review could be improved in two ways: it should cite original references as much as possible and describe what is currently known on several aspects with more accuracy, as described in the following points.

 Answer: Thank you for your comments, we have modified the text as suggested.

  1. Introduction:

Reviewer comment: "Ras-like as molecular switches". Please cite the original work of Bourne, Sanders and McCormick, published in 1992, much earlier than the proposed references (10.1038/348125a0)

 Answer: This reference has been modified as suggested.

  1. Rho GTPases regulation, family members, and roles.

Reviewer comment: Li 87-88: " The Rho GTPase family is composed of 21 proteins with almost ubiquitous expression ..."

The Rho family of GTPases is actually made of 20 proteins, not 21. This was established by Boureux et al. 2007 (DOI: 10.1093/molbev/msl145). In this reference, it is also established that nine of the 20 members show a marked tissue-specific expression. The sentence should be modified accordingly and the reference should be added.

Answer: This reference has been modified as suggested. “The Rho GTPase family is composed of 20 proteins. Most of them exhibit an almost ubiquitous expression, but some (e.g. RhoH, Rac2) have an evident tissue-specific distribution [37]”.

Reviewer comment: Li 93-94: The sentence "a small subset of Rho GTPases (e.g., RhoE) is characterized by being present in a constitutive active state" is not correct, since at least eight Rho members (i.e. 40%, not a "small subset") are either devoid of GTPase activity or exchange spontaneously GDP with GTP. This has been reviewed in Aspenström et al. 2007 (10.1016/j.yexcr.2007.07.022). The sentence should be modified accordingly and the reference should be added.

Answer: Thank you for your comment. The text has been modified as suggested.

Reviewer comment:  Li 101: " skeletal muscle-specific functions (ArhGEF3, Dock1, Dock5, Trio, Vav2)

The authors should also add to the list Obscurin, involved in skeletal muscle functions, as described li 263-272, and ARHGEF25/GEFT (Bryan et al. 2005, 10.1128/MCB.25.24.11089-11101.2005).

Answer: Thank you for noticing this error, this has now been modified as suggested.

Reviewer comment:  Li 115-116: The authors cite the ref #45 to state that the human genome encodes 85 RhoGEFs. However, the number of Dbl-like members had been miscounted in this reference, which stated 74 in the legend whereas only 70 were listed in the table. Since then, the human repertoire of Dbl-like proteins has been worked out extensively in Fort and Blangy, 2017 (DOI: 10.1093/gbe/evx100), which identified 71 Dbl-like genes in the human genome. In addition, SWAP70 (Shinohara et al. 2002, 10.1038/416759a) and SWAP70L/DEF6 (Gupta et al. 2003, 10.1016/s0198-8859(03)00024-7) have been considered as Dbl-like RhoGEFs, although their exchange domains do not have the conserved residues found in the DH domains of other Dbl-like members. The RhoGEF family in humans is thus made of 11 DOCK genes, 71 Dbl-like and 2 SWAP70. The sentence should be modified accordingly and the reference should be added.

Answer: Thank you for noticing this. The text has now been modified and the new reference included.

Reviewer comment:   Li 131: The authors indicate that "more than 70 effectors have been identified". As worked out by Bagci et al. 2020 (DOI: 10.1038/s41556-019-0438-7), 239 effectors have been identified in only two human cell types (HeLa and HEK293T cells). The sentence should be modified accordingly and the reference should be added.

 Answer: The text has now been modified and the new reference included.

2.1 Rho GTPases in muscle differentiation:

Reviewer comment:  Li 191: Ref #39 (Laurin et al. 2008) is not at the correct place. Since it only concerns Dock1‑/‑ mice, it should be placed at the beginning of the line above, after "animals".

 Answer: Thank you for noticing this issue, this has now been modified as suggested.

Reviewer comment:  TC10 should be added to the list of GTPases involved in differentiation, as mentioned in section §2.4.

 Answer: Given that we already had a specific section for myofibrillogenesis and that we have decided to focus the discussion on this section on earlier steps up to myotubes, we consider appropriate to maintain all the TC10-related information in the section 2.4.

Reviewer comment: DEF-6 also influences myoblast differentiation (Samson et al. 2007, 10.1074/jbc.M611197200). The reference should be added.

Answer: There are some issues regarding this paper. The first one is that the activation pattern of Rac1 during differentiation is not consistent with what is known about the positive role of Rac1 in myoblast fusion and what it has been published using the same cell line (see Ref. 44 and 78, for example). The second issue is that all their results are based on ectopic overexpression of Def6, with levels much higher than the physiological ones that may give an artefactual activation of Rac1. Moreover, Def6 deficient mice do not seem to have muscle defects as reported for Trio and Dock1 deficient mice We have nonetheless included this reference in the text and figure as indicated by the reviewer.

2.2 Rho GTPases in muscle regeneration:

Reviewer comment:  In addition to ARHGEF3, the authors should add ARHGEF25, shown to play roles in myogenesis and muscle regeneration (GEFT, Bryan et al.2005, 10.1128/MCB.25.24.11089-11101.2005)

Answer: Thank you for your input, we have now included this reference in the text.

2.3 Rho GTPases in muscle mass regulation:

Reviewer comment:  This section reports on skeletal muscle mass homeostasis, with a particular emphasis on Vav2 knock-in mice (Ref #43), as recently published by the authors.

Since the review published in Cells in 2019 (Ref #26) overlaps some aspects developed in this section with more details, it is needed to state that this section summarizes recent work on Vav2 and specify that the roles of other components of Rho GTPase signaling pathways in exercise and glucose uptake have been reviewed in Ref  #26.

 Answer: Thank you for your suggestion. We feel that our section does not overlap with that review, since the focus is different and we are only including results obtained from mouse models and none of the correlational evidences referenced in the 2019 review. Moreover, we feel that the comparison of the defects observed in the Pak and Vav2 mouse models enriches the discussion and offers a different point of view.

We have now indicated that additional information can be found in the Cells, 2019 review.

2.4 Rho signalling in sarcomere banding pattern:

Additional information could be included in this very short section:

Reviewer comment:  Benian et al. 1996 (10.1083/jcb.132.5.835 ) identified the presence of DH/PH domains in unc-89, the C. elegans ortholog of obscurin. Young et al. 2001 (ref #94) also showed the presence of the DH/PH domains but did not show that obscurin is a RhoGEF (i.e. capable of GDP/GTP exchange activity). This was done in Qadota et al. 2008 (10.1016/j.jmb.2008.08.083), showing that the DH/PH domain of unc-89 has an exchange activity toward Rho, not Rac, Cdc42 or Mig-2. Note also that O Raeker et al. 2010 (10.1016/j.ydbio.2009.11.018) showed the direct implication of the RhoGEF domain in the physiological functions of obscurin in the zebrafish, by using a morpholino-mediated in-frame deletion approach.

Answer: Thank you for your suggestion. Although this information is very interesting, we consider that it is not necessary to understand the role of obscurin in skeletal muscle. Additionally, this level of detail has not been included for any of the other Rho GEFs mentioned in the text.

Reviewer comment:  Last, obscurin is considered as a potential therapeutic target in muscle disorders (Randazzo et al. 2017, 10.1080/14728222.2017.1361931; Perry et al. 2013, 10.1002/iub.1157). This can serve as a link between this section and the next one.

References should be added and the text modified accordingly.

 Answer: Thank you for your comment, this information has been now included.

2.5 Rho GTPases in skeletal muscle-related disease.

Additional references should be included and the text modified accordingly:

Reviewer comment:  In stretch or contraction stimulated muscles, glucose uptake requires Rac1 activity (Sylow et al. 2014, 10.1113/expphysiol.2014.079194 and Sylow et al. 2015, 10.1113/jphysiol.2014.284281). Tiam1 may be one RhoGEFs responsible for Rac1 activation in this process (Yue et al. 2021, 10.1096/fj.202001312R; You et al. 2013, 10.1016/j.cellsig.2013.08.018).

The RhoGEFs  Sos (Kim et al. 1999, 10.1006/bbrc.1998.9940), Kalirin (Lee et al. 2017, 10.1016/j.cellsig.2016.10.013), SWAP70 (Ueda et al. 2008, 10.1042/BC20070160) have been also involved in glucose uptake.

Answer: Thank you for your suggestions. Since the paragraph is referring to type II diabetes and insulin signalling, and this information has already been summarised in the review by Machin et. Al (Cells, 2021; https://doi.org/10.3390/cells10040915) that we reference in the text, we did not include this information in the first version of the manuscript.

The involvement of Sos in insulin signalling is Ras-dependent and Rho-GTPase independent, as indicated in the proposed reference and others (e.g. this review by R. Kahn: PMCID: PMC3941218). Kalirin has been shown to be involved in glucose uptake only in response to the myokine follistatin-like protein 1, but the relevance of this process in whole body homeostasis has not been established. Moreover, as stated in the review by Machin et al (https://doi.org/10.3390/cells10040915), the concentrations used are at least of an order of magnitude higher than the physiological ones. Regarding the last reference, we think the reviewer is referring to Plekhg4 (FLJ00068) and not SWAP70. In this case, again, we decided to keep this information out of the paper in the first version since it have been previously revised in the two reviews referenced in the text.

We have now included some of these references in the text (lines 319-322): “The Rho GEF Plekhg4 has been shown to promote Rac1 activation in this process [114]. Rac1 and the Rho GEFs Tiam1 and Kalirin have also been shown to mediate muscle glucose uptake in response to other stimuli [115-119].”

Reviewer comment:  Last, GRAF1/ARHGAP26 is necessary for efficient muscle membrane repair and may be involved in the pathological progression of muscular dystrophy patients  (Lenhart et al. 2015, 10.1186/s13395-015-0054-6).

 Answer: We have now modified the text accordingly.

Figures:

Reviewer comment:  Figures should be modified according to new references included in the text.

Answer: We have now modified the figures accordingly.

Reviewer comment:  More contrast should be added as some parts are difficult to read.

Answer: Thank you for letting us know. We have now modified the pictures as suggested.

Reviewer 2 Report

In this review manuscript, the authors have surveyed the literature regarding the role of Rho GTPases in skeletal muscle development and homeostasis. For the most part, the review is clearly written and informative. It will be a useful resource for researchers working both in Rho GTPase signaling and muscle. The manuscript could use some more proofreading for clarity and style. In addition, the figures in the manuscript version sent to reviewers were pixelated and unclear, this should be improved. Below are a number of suggestions that should be considered in a revised version.

  1. Line 28 -strictly speaking the skin is the largest organ in humans.
  2. There are numerous examples where proof-reading would help to improve the clarity of the manuscript. The first glaring example is on line 29 “given its extension” isn’t clear, do you mean to say “By extension…” Another is on line 38 “other life periods”, what does actually mean?
  3. Line 103 – although the majority of Rho GTPases are geranylgeranylated, some are farnesylated and some may be modified by either form of prenylation. For example, this paper reported the preferences for many Rho GTPases https://www.ncbi.nlm.nih.gov/pmc/articles/PMC2533093/
  4. Lines 103-104. In addition to prenylation, palmitoylation plays roles in membrane localization of Rho GTPases. This review is a good summary. https://www.sciencedirect.com/science/article/pii/S0962892412000670?via%3Dihub
  5. Line 140 – extra “cells”
  6. Some of the figures that have been imported into the manuscript are quite pixelated. The final submitted form should replace these figures with higher resolution versions.
  7. It’s not clear what “incrementing” autophagy in mice on line 221 means. Incrementally increasing autophagy?
  8. Lines 251-252 – It’s hard to understand how Rac1 works downstream of VAV2 in an IGF1-Akt-mTOR pathway, when the previous sentence only describes that varyingly active forms of VAV2 have differential effects on muscle mass. Can this be explained more thoroughly?
  9. Line 312 – It would be helpful to say that the DMD gene encodes for the dystrophin protein.

Author Response

Reviewer comment: In this review manuscript, the authors have surveyed the literature regarding the role of Rho GTPases in skeletal muscle development and homeostasis. For the most part, the review is clearly written and informative. It will be a useful resource for researchers working both in Rho GTPase signaling and muscle. The manuscript could use some more proofreading for clarity and style. In addition, the figures in the manuscript version sent to reviewers were pixelated and unclear, this should be improved. Below are a number of suggestions that should be considered in a revised version.

Answer: Thank you for your kind comments, we appreciate you consider our review as a useful resource for the field.

Reviewer comment: Line 28 -strictly speaking the skin is the largest organ in humans.

Answer: If we consider skeletal muscle as a whole, the sum of their weight and volume is larger than skin. However, to avoid any confusion, we have modified this sentence in the new version of the manuscript.

Reviewer comment: There are numerous examples where proof-reading would help to improve the clarity of the manuscript. The first glaring example is on line 29 “given its extension” isn’t clear, do you mean to say “By extension…” Another is on line 38 “other life periods”, what does actually mean?

Answer: Thank you for detecting these issues. We have now revised the manuscript and corrected the indicated phrases.

Reviewer comment: Line 103 – although the majority of Rho GTPases are geranylgeranylated, some are farnesylated and some may be modified by either form of prenylation. For example, this paper reported the preferences for many Rho GTPases https://www.ncbi.nlm.nih.gov/pmc/articles/PMC2533093/

Answer: Thank you for your indication, we have now indicated prenylation instead of geranyl-geranylation (see line 103).

Reviewer comment: Lines 103-104. In addition to prenylation, palmitoylation plays roles in membrane localization of Rho GTPases. This review is a good summary. https://www.sciencedirect.com/science/article/pii/S0962892412000670?via%3Dihub

Answer: Thank you for your comment, we have now included this additional modification (see line 104).

Reviewer comment: Line 140 – extra “cells”

Answer: Thanks for detecting this typo, it has now been corrected.

Reviewer comment: Some of the figures that have been imported into the manuscript are quite pixelated. The final submitted form should replace these figures with higher resolution versions.

Answer: Thank you for letting us know. The quality of the images has been changed during the editorial process, but we will ensure they can be clearly read in the revised version.

Reviewer comment: It’s not clear what “incrementing” autophagy in mice on line 221 means. Incrementally increasing autophagy?

Answer: Thank you noticing this mistake, we were referring to an increase in autophagy. This has now been corrected in the text.

Reviewer comment: Lines 251-252 – It’s hard to understand how Rac1 works downstream of VAV2 in an IGF1-Akt-mTOR pathway, when the previous sentence only describes that varyingly active forms of VAV2 have differential effects on muscle mass. Can this be explained more thoroughly?

Answer: We have now modified this paragraph to make it clearer: We have recently shown that the expression of a catalytically hyperactive or a hypo-morphic Vav2 mutant version from the endogenous Vav2 locus promotes increased and reduced lean mass, respectively, in knock-in mouse models [46]. Further experiments indicate that the Rho GEF activity of Vav2 contributes to the activation of the IGF1 signaling pathway, being Rac1 the Rho GTPase activated by Vav2 in this new regulatory layer of the IGF1-Akt-mTOR pathway [46]..

Reviewer comment: Line 312 – It would be helpful to say that the DMD gene encodes for the dystrophin protein.

Answer: We have now included the name of the protein in line 334: “This gene encodes a protein called dystrophin that forms complexes with extracellular and transmembrane glycoproteins.”